# Lipid Profiles of Human Milk and Infant Formulas: A Comparative Lipidomics Study

**DOI:** 10.3390/foods12030600

**Published:** 2023-02-01

**Authors:** Danjie Wu, Le Zhang, Yan Zhang, Jiachen Shi, Chin Ping Tan, Zhaojun Zheng, Yuanfa Liu

**Affiliations:** 1State Key Laboratory of Food Science and Technology, School of Food Science and Technology, National Engineering Research Center for Functional Food, National Engineering Laboratory for Cereal Fermentation Technology, Collaborative Innovation Center of Food Safety and Quality Control in Jiangsu Province, Jiangnan University, Wuxi 214122, China; 2Department of Neonatology, The Affiliated Wuxi Children’s Hospital of Nanjing Medical University, Wuxi 214023, China; 3Department of Food Technology, Faculty of Food Science and Technology, University Putra Malaysia, Serdang 43400, Malaysia

**Keywords:** human milk, infant formula, lipid profiles, lipidomics, significantly differential lipids

## Abstract

Infant formulas (IFs) are prevalent alternatives for human milk (HM), although their comparative lipid profiles have not been fully investigated. We adopted lipidomics to analyze and compare in-depth the lipid patterns of HM and IFs. The results indicated that the distribution of fatty acids (FAs) and the structure of triacylglycerols varied substantially in the analyzed samples. A total number of 425 species were identified during the analysis. HM was abundant in triacylglycerols that contained unsaturated and long-chain FAs (>C13), while triacylglycerols in IFs were mainly comprised of saturated and medium-chain FAs (C8−C13). Higher levels of sphingomyelin were observed in HM. Furthermore, HM and IF1 contained 67 significantly differential lipids (SDLs), and 73 were identified between HM and IF2. These SDLs were closely associated with nine metabolic pathways, of which the most significant was the glycerophospholipid metabolism. The results shed light on the differences between the lipid profiles of human and infant formula milks, and provide support for designing Chinese infant formula.

## 1. Introduction

Human milk (HM) is universally acknowledged as the optimal nutrient source for neonates and infants. As one of the major constituents of HM, lipids account for 3−5% of its substance, and participate in critical biological processes such as cell membrane formation and signal transmission, as well as providing more than 50% of the energy, essential fatty acids (FAs), and fat-soluble vitamins to meet nutritional requirements [1,2]. Lipids exist in the milk duct in the form of milk fat globule (MFG), composed of a 98% triacylglycerol (TG) core and a trilaminar membrane called the milk fat globule membrane (MFGM). This membrane contains phospholipids (PLs), cholesterols, proteins, glycoproteins, gangliosides and enzymes [3]. Compared to other mammalian milks, HM has a unique FA distribution, consisting of more than 60−70% palmitic (P), mainly esterified at the *sn*-2 position, while most unsaturated fatty acids (UFAs) are generally situated at the *sn*-1 or *sn*-3 position of the glycerol backbone [4]. According to previous research, the preferential esterification of FAs aids in the absorption of FAs and calcium, which perform critical functions in neonatal development [5]. 

Infant formulas (IFs) are usually the first choice when breast-feeding is unavailable due to health-related issues or unaccommodating work environments [6,7]. However, it is still questionable whether the nutritional levels of IFs are comparable with that of HM, particularly the highly diversified lipids. HM and IFs have been widely investigated by comparing their similarity and difference of the lipid profiles. For instance, Sun et al. [8] identified 29–40 TGs from 180 commercial formulas and 29 TGs from mature milk using non-aqueous reverse-phase high-performance liquid chromatography (NARP-HPLC), demonstrating the significant differences of TG composition and the abundance of PA at the *sn*-2 position among HM and IFs. Chinese HM samples at different lactation stages were also found to obviously differ in TG composition from those of IFs [9,10]. Milk fat contains several thousand lipid species, which is regarded as one of the most complicated substances materials in nature [7]. The above-mentioned studies are limited to the TG composition between HM and IFs, while the comparative analysis in-depth of total lipids was underestimated. Therefore, finding a novel approach to evaluate the lipid profiles is indispensable for the comprehensive analysis of HM and IFs.

Lipidomics, an important branch of metabolomics, has become a rapidly developing technology and is widely used for comprehensive analysis of the lipidomes in various biological samples [7,11,12,13]. Recently, a high degree of attention has been paid to the lipidomes in HM and/or IFs using lipidomics studies. Lipidomics was adopted to perform comparative analysis of lipid composition from different mammalian sources of milk, presenting a total of 13 lipid classes [7]. In addition to the TGs, glycerolipids (GLs), glycerophospholipids (GPs) and sphingolipids (SPs) were observed in Chinese HM [14]. Zhang et al. [15] also analyzed the lipidome of HM and IFs using UHPLC-Q-TOF-MS, and identified 48 and 71 common lipid species, respectively, using positive (POS) and negative (NEG) ion modes. Numerous studies put emphasis on the TG composition of milk lipids, whereas the total lipid profiles (TGs, PLs, cholesterols, etc.) among HM and IFs have yet to be explored [1,9,16,17]. To our knowledge, comprehensive reports on the comparison of HM and IF lipidome are seldom reported. Consequently, the present research aims at comparing the lipid profiles of HM and IFs by means of lipidomics. Additionally, the common and differential components among the two samples were identified to illustrate the lipid patterns in HM and IFs. Furthermore, the metabolic pathways in which these differential lipids are involved were analyzed by mapping to the KEGG, HMDB, and PubChem databases. Overall, the present study has the potential to serve as a guide in the development of IFs and related products for Chinese babies. 

## 2. Materials and Methods

### 2.1. Sample Collection

Six mature milk samples (collected from 30 to 32 days) were kindly obtained from healthy volunteer mothers of similar ages in the Wuxi Children’s Hospital. All the volunteers were residents of Wuxi and similar dietary patterns were ensured. All samples were collected after infant feeding and were repacked and stored at −80 ℃ before individual analysis. All volunteers received detailed information about the study, and the protocol was approved by the Ethics Committees of Jiangnan University and Wuxi Children’s Hospital. The two infant formulas are expressed as IF1 (milk-based) and IF2 (plant-based). Generally, IFs were dissolved in water at a ratio of 1:10 before analysis as reported [18]. Detailed information about the sources of fat in IFs is shown in Appendix A. 

### 2.2. Chemicals

The internal standards (d5-17:0/17:1/17:0) TG, 15:0 lysophosphatidylcholine (LPC), (19:0/19:0) phosphatidylethanolamine (PE), 13:0 lysophosphatidylethanolamine (LPE), d18:1/6:0 sphingomyelin (SM), (17:0/17:0) phosphatidylserine (PS), and (17:0/17:0) phosphatidylglycerol (PG) were purchased from Avanti Polar Lipids, Inc. (Alabaster, AL, USA, 99%). A fifty-two fatty acid methyl ester (FAME) solution was obtained from NU-CHEK-PREP Co., Ltd. (Elysian, MN, USA, ≥99%). Porcine pancreatin (4 × USP, CAS: 8049-47-6) was purchased from Sigma-Aldrich (St. Louis, MO, USA). Cholesterol standard (CAS: 57-88-5, ≥99.5%) was obtained from Beijing Dingguo Changsheng Biotechnology Co., Ltd. (Beijing, China). Sodium cholate (CAS: 361-09-1, 98%) was obtained from J&K Scientific Ltd. (Beijing, China). Silica gel G TLC plate was purchased from Bangkai (Jining, China). All analytical solvents (methyl tert-butyl ether (MTBE), CHCl_3,_ CH_3_OH, n-hexane, etc.) used herein were high performance liquid chromatography or analytical reagent grade.

### 2.3. Lipid Extraction for GC and UHPLC-Q-TOF-MS

Total lipids were extracted from HM and IFs according to the method of Zhang et al. [19]. HM samples were mixed with CHCl_3_/CH_3_OH (2:1, *v*/*v*) at a ratio of 1:3, followed by 30 min of ultrasonic treatment at 30 °C. After 10 min of centrifugation at 16,994× *g* and 4 °C, the organic phase was transferred to a new vial, and the same extraction process was performed on the remaining water phase once more. The mixed organic phase was evaporated to about 5 mL under vacuum, and further dried under a stream of nitrogen. IFs were dissolved in ultrapure water at a ratio of 1:10 before mixing with CHCl_3_/CH_3_OH (2:1, *v*/*v*). The obtained dry lipid was stored at −20 °C for further gas chromatography (GC) analysis.

The process for lipid extraction was performed following the method of Li et al. [12], with slight modifications. Briefly, 60 μL milk sample and 340 μL ultrapure water were collected and placed in an EP tube (A). Then, 960 μL extraction liquid (MTBE:CH_3_OH = 5:1, *v*/*v*) and 70 μL internal standard mixture ((d5-17:0/17:1/17:0) TG, 15:0 lyso PC, (19:0/19:0) PE, 13:0 lyso PE, d18:1/6:0 SM, (17:0/17:0) PS, and (17:0/17:0) PG were mixed in equal quantities in the concentration of 250 μg/mL) were added to each sample. The sample was mixed to homogeneity, vortexed, sonicated for 10 min, and then centrifuged for 10 min at 16,994× *g* and 4 °C. The supernatant was transferred to a new EP tube (B), and the remaining aqueous phase was extracted again with extraction liquid. The organic phases were combined and dried in a vacuum concentrator at 30 °C before 100 μL isopropyl alcohol was added for reconstitution. The samples were then transferred to a fresh glass vial for ultra-high-performance liquid chromatography quadruple time-of flight mass spectrometry (UHPLC-Q-TOF-MS) analysis.

### 2.4. Determination of Lipid and Cholesterol Content

Lipid content was determined using gravimetric method after being extracted by CHCl_3_/CH_3_OH (2:1, *v*/*v*).

Cholesterol content was measured according to Zhang et al. [19]. Briefly, milk samples were first saponified in ethanol and KOH solution and then extracted with ether/petroleum ether. The solvent was dried under vacuum and the residue was redissolved in absolute ethanol, followed by HPLC (Shimadzu, Kyoto, Japan) with a C18 column (4.6 × 250 mm × 5 μm, Bonna-Agela, Tianjin, China), equipped with an SPD-20A UV/VIS detector. Methanol was selected as mobile phase and delivered at 1.0 mL min^−1^. The injection volume and column temperature were set at 10 μL and 40 °C, respectively. Cholesterol was determined at 205 nm and quantified by using the external standard method.

### 2.5. Determination of Total Fatty Acids

Analysis of total FAs was completed using a GC system (Nexis GC-2030, Shimadzu, Kyoto, Japan) equipped with a capillary column (60 m × 0.25 mm × 0.25 μm), as described by Ye et al. [20]. Briefly, milk lipids extracted previously were mixed successively with KOH-CH_3_OH (0.5 M, 2 mL), incubated for 30 min with shaking at 65 °C, and BF_3_-CH_3_OH (1:3, *v*/*v*, 2 mL), and then incubated for 10 min at 70 °C. The generated FAMEs were extracted with 2 mL n-hexane (HPLC grade) and then through a 0.22 μm organic filter membrane before GC analysis. The split ratio was 20 and the carrier gas flow rate was 1.8 mL/min with 1 μL of injection volume. The identification of FAs was accomplished by matching the retention time with that of FAME standards.

### 2.6. Determination of sn-2 Fatty Acids

The determination method of *sn*-2 FAs was as follows. Firstly, TG were hydrolyzed by the method of Sun et al. [21]. Then, the hydrolytic product was separated on a silica gel G TLC plate and the hexane/diethyl ether/acetic acid (50:50:1, *v*/*v*/*v*) acted as developing solvents. The band corresponding to 2-monoglyceride (MG) was scraped off and extracted twice with diethyl ether (1 mL). After that, the extracted liquid was evaporated under nitrogen gas and the residue was methylated, as in Section 2.5, before GC analysis.

### 2.7. Determination of Lipid Profiles

Lipid analysis was performed by the method of Li et al. [12], with some modifications of the equipment. This process was completed using an UHPLC system with a Phenomenex Kinetex C18 column (2.1 × 100 mm × 2.6 μm, Phenomenex, Los Angeles, CA, USA) coupled with a Triple TOF 5600 (Q-TOF; AB Sciex, Framingham, MA, USA). The mobile phase elution gradient was as follows: 0 min, 40% B; 12 min, 100% B; 13.5 min, 100% B; 13.7 min, 40% B; 18 min, 40% B, where phase A was 10 mM ammonium acetate + 40% H_2_O + 60% acetonitrile, and phase B was 10 mM ammonium acetate + 10% acetonitrile + 90% isopropanol. The flow rate was 0.3 mL/min. The injection volumes were 2 and 6 μL for POS and NEG ion modes, respectively.

The ESI source was used in this determination and the conditions were set as follows: spray voltages of 5500 V and −4500 V in positive and negative modes, respectively; source temperature of 550 °C; atomizing air pressure of 60 Pa and curtain pressure of 30 Pa; declustering potential of 100 V. The data with mass range of m/z 50–1300 was acquired and the top 10 precursor ions were chosen for fragmentation. The identification of lipid classes was achieved using the exact mass of the molecular ion plus characteristic fragmentation, and the quantification of each class was completed using the added internal standard. GLs were detected in POS ion mode with the adduct ion of “+NH_4_”, and identified based on the m/z of the residue losing a fatty acyl. PLs were detected in POS or NEG ion modes with the adduct ion of “+H” or “−H” and “+CH3COO” respectively, and qualitatively analyzed based on the fatty acyl residues directly. The lipid content was equal to the ratio of the extracted peak area to the internal standard area times the internal standard concentration. Then, the content of each lipid was normalized in POS and NEG ion modes, respectively.

### 2.8. Statistical Analysis

In this study, each HM was analyzed separately as biological duplication, while the infant formulas were tested in duplicate. The qualitative analyses of GLs and PLs were accomplished using MSDIAL 3.0.0.0 combined with Global Natural Product Social Molecular Networking. The variable importance in projection (VIP) value was performed using the SIMCA 14.1 software package (Copyright ©1998–2015 MKS Umetrics AB). Hierarchical clustering analysis was performed using https://www.omicstudio.cn/tool (accessed on 3 May 2022). The correlation network analysis was completed using Cytoscape 3.9.0. Lipid metabolism pathway analysis was performed using the Web-based platform MetaboAnalyst (https://www.metaboanalyst.ca/, accessed on 10 May 2022) [22]. The results were expressed as means (%) ± standard deviations (SD), and significance analysis were performed using the JMP Pro 14 (SAS Institute Inc., Cary, NC, USA).

## 3. Results and Discussion

### 3.1. Total Lipid Comparation

The total lipid content observed in HM was 43.90 ± 4.60 g/L (Table 1), corresponding to the previous findings (43.57 ± 6.05 g/L (*n* = 103), 4.79 ± 1.71 g/100 mL (*n* = 12)) [16,23]. Comparatively, IFs diluted to the recommended concentration showed the lipid levels of approximately 20.00 g/L, which were roughly two times lower than that in HM. Notably, lower lipid contents in IFs may favor the supply of reduced concentrations of essential FAs and provide lower caloric intake [24].

As one of the fundamental components in milk lipids, cholesterol serves as not only a precursor of vital biomolecules like steroids hormones, but also an important constituent in cell membranes and lipoproteins. As depicted in Table 1, the cholesterol level was found to be 12.14 ± 3.09 mg per 100 g HM, which was in agreement with the findings about the breast milk (13.1 mg/100 g, 13.81 mg/100 g) [25,26]. By contrast, IFs presented a low cholesterol content, which only accounted for approximately one sixth of that in HM. Also, IF1 had a 1.5-fold higher cholesterol content than IF2, partly due to the addition of raw bovine milk. It is beneficial for infants to consume a high level of cholesterol, which greatly lessens the risk of cardiovascular disease in adult life via regulation of long-term cholesterol metabolism [26]. In addition, the proportion of cholesterol to total lipids was much higher in HM than that in IFs. This proportion may be attributed to the nutritional properties of HM, corresponding to the view that breast-feeding (particularly when exclusive) during early infant life may be associated with lower blood cholesterol concentrations in adulthood [27].

### 3.2. Composition of Total Fatty Acids

The composition of total FAs in HM and IFs is shown in Table 2. Thirty-five fatty acids were detected in HM and IFs in total, including sixteen saturated fatty acids (SFAs), ten monounsaturated fatty acids (MUFAs), and nine polyunsaturated fatty acids (PUFAs). In HM, MUFAs (38.97%) and SFAs (38.15%) were the most abundant FAs, followed by PUFAs (22.84%). In contrast to the present study, Zhang et al. [15] revealed that PUFAs comprised the most species, followed by SFAs and MUFAs. Bah et al. [28] reviewed the FA composition of breast milk in various populations and demonstrated that these three types of FAs on the basis of proportions were ordered as SFAs > MUFAs > PUFAs. Such slight differences may be attributed to various factors, such as interindividual variation, maternal dietary intake, geographic region, cultural differences, socio-economic status, and so on [28]. In IF1, SFAs (41.28%) were the most abundant, followed by MUFAs (36.94%) and PUFAs (21.76%). In contrast to IF1, MUFAs (46.34%) took the first place, followed by SFAs (32.07%) and PUFAs (21.58%) in IF2. The different fat sources between IF1 and IF2 could be responsible for their contrary proportions of MUFAs and SFAs. 

Interestingly, caproic acid (Co, C6:0) was the only short-chain FA (SCFA) observed in HM and IF1, while no SCFA was detected in IF2. In terms of MCFAs, HM presented the level of 5.81%, which was over twice as much in IF1 (2.61%), but only accounted for approximately half of IF2 (*p* < 0.05). This quantitative difference might be explained by the varied fat sources, of which one contains plentiful of MCFAs. MCFAs were evidenced to be essential for infant growth, along with a more rapid and efficient absorption compared to that of LCFAs [29]. Specifically, the most prevalent MCFAs were capric acid (Ca, C10:0) and lauric acid (La, C12:0), which accounted for 5.59%, 2.3%, and 11.65% in HM, IF1 and IF2, respectively. As expected, P was the most prevalent SFA in HM with the proportion of 20.92%, as well as in IF1 (29.21%), which were two or three times higher than that in IF2 (10.08%). The highest content of P in IF1 might be at least partially due to the supplemental addition of 1,3-Dioleoyl-2-palmitoylglycerol (OPO). Stearic acid (St, C18:0), the second most abundant SFA, showed the highest level of 6.05% in HM, followed by 5.89% and 4.31% in IF1 and IF2, respectively. 

As for the MUFAs, HM showed the level of 38.97%, which was consistent with IF1 (*p* > 0.05). Compared with these two samples, IF2 presented much higher content of MUFAs (*p* < 0.05). Specifically, oleic acid (OA, C18:1) was the most prevalent type in all milk lipids analyzed, corresponding to the previous reports about human milk and infant formula based on mammal milk [30,31,32]. Comparatively speaking, the almost identical levels of OA were observed in HM (35.56%) and IF1 (35.93%), which were significantly lower than that in IF2 with the value of 45.64% (*p* < 0.05). It is worth noting that to better mimic HM, the proportion of vegetable oils with a high OA content should be reduced in IF2. Another important MUFA was nervonic acid (NA, C24:1), which plays a crucial role in the development of central nervous system myelination although no NA was detected in formulas in this study [33]. As proposed, the level of NA in IFs should be increased to ameliorate their nutritional value and achieve a higher similarity with HM [34]. 

Among all the investigated milk lipids, no significant difference was observed in PUFAs, accounting for the percentage of approximately 22%. Linoleic acid (LA, C18:2) was the predominant PUFA in all samples analyzed and the level detected here was in the range of other studies [4,35]. In addition, HM showed higher levels of α-linolenic acid (ALN, C18:3, 0.51%), arachidonic acid (ARA, C20:4, 0.11%), and docosahexaenoic acid (DHA, C22:6, 0.27%) than IFs. The latter two species belong to LCPUFAs, which play an important role in the development of cognitive and visual functions in infants, thereby influencing their later childhood [36]. Although ARA and DHA could be supplemented by dietary intake or synthesized from LA and LN under the action of desaturases and elongases [37], the immature enzymes in infants largely limit this transformation. Accordingly, the increasing levels of these two acids should be considered when designing infant formula. 

### 3.3. Composition of sn-2 Fatty Acids

The composition of *sn*-2 FAs in HM and IFs is shown in Table 3. We detected 25, 10 and 13 types of *sn*-2 FAs in HM, IF1 and IF2, respectively. Clearly, the *sn*-2 FAs of HM were more complex than those of formulas. According to the degree of unsaturation, they were separated into three groups, including SFAs, MUFAs and PUFAs.

The percentage of total SFAs that occupied the *sn*-2 position was over 75% in HM, while all types of formulas had significantly lower levels (*p* < 0.05). More specifically, the SFAs concentration in HM (76.65%) was 3.3 times higher than in IF2 (23.24%). Especially of note, P accounted for 55% at the *sn*-2 position in HM, presenting the absolute *sn*-2 selectivity. Compared to HM, IF1 containing raw bovine milk showed relatively lower content of *sn*-2 P (42.95%), but this was over 14-fold higher than that in IF2, which was based on plant oils. Bovine milk probably provided suitable levels of P at the *sn*-2 position to better mimic the positional selectivity of this FA. Other SFAs, with the exception of C18:0, also had a specific *sn*-2 position distribution in HM (Figure 1). This preferential distribution pattern of SFAs in HM was distinguished from the formulas, further confirming the findings described by other researchers [21,30,38]. Specifically, higher levels of SFAs at the *sn*-2 position are indispensable to increase the absorption of FAs and calcium, as well as improve bone matrix quality and stool consistency for formulas [39]. 

The *sn*-2 positional content of total MUFAs was 10.67% in HM, which was two or five times lower than that in IF1 (24.22%) and in IF2 (52.72%). As observed, *sn*-2 MUFAs of formulas were related to the type of fat added, presenting much higher *sn*-2 MUFA levels in plant-based formulas (IF2) than that in milk-based one (IF1). This result was in accordance with the findings of Sun et al. [21] on the *sn-2* fatty acid composition in commercial infant formulas with different fat sources, in which the *sn*-2 MUFA level in plant-oil formula (37.73%) was higher than those in cow’s milk formula (31.31%) and goat’s formula (31.05%).

The level of PUFAs presented at the *sn*-2 position in HM was 12.31%, slightly lower than that in IF1(17.93%) and two times lower compared to IF2 (24.05%). In agreement with Sun et al. [21], most of LCPUFAs in formulas were not observed at the *sn*-2 position. As shown in Table 3, PUFAs located at the *sn*-2 position were composed of L and LN in all investigated milk samples, accounting for more than 78% of total PUFAs. In addition, LCPUFAs were observed therein, presenting as ARA, mainly located at the *sn*-2 position in HM. DHA was distributed randomly in HM and preferentially located at the *sn*-2 position in IF2. Similarly, Deng et al. [38] also found that most LCPUFAs were acylated in the *sn*-2 position by analyzing the distribution of FAs in HM from different lactation stages and geographic locations/cities.

### 3.4. Comparative Analysis of Lipids

#### 3.4.1. Qualitative Analysis of Lipids

Lipids were detected both in POS and NEG ion modes in order to obtain greater numbers of lipids. Three categories, including GLs, GPs and SPs, 15 subclasses of 425 species of lipids were identified in total after filtering out lipids detected in both ion modes. The number of identified lipid species in studied samples using UHPLC-Q-TOF-MS in present work were superior to those in human milk using SFC coupled with Q-TOF-MS and donkey milk lipids via quantitative lipidomics [9,12]. Among the three lipid categories, there were three subclasses of GLs (TGs, diglycerides (DGs), and MGs), eight subclasses of GPs (phosphatidylcholine [PC], PE, LPC, LPE, PS, phosphatidylinositol [PI], phosphatidic acid [PA], and PG), and four subclass of SPs (SM, ceramide [Cer], hexosylceramide [Hexcer], and dihexosylceramide [Hex2cer]) (Figure 2A). The number of TGs species constituted the most of total species identified here. In POS ion mode, 291 different lipids were identified, including 214 GLs (156 TGs, 46 DGs, and 12 MGs), 60 GPs (14 PCs, 29 PEs, 2 LPCs, and 15 PGs), 17 SPs (6 SMs, 2 Cers, 3 Hexcer, and 6 Hex2cers) (Figure 2B). Among these lipids, 222 species of lipids (154 TGs, 46 DGs, 9 MGs, 7 PCs, 2 PEs, 1 LPC, 1 PG, 1 SM, and 1 Cer) were common to all investigated milk samples. It can be concluded that GPs, especially PEs, contribute greatly to the differences between HM and formulas. In addition, a total of 134 lipids were identified in NEG ion mode, consisting of 85 GPs (thirty-one PEs, six LPCs, five LPEs, twelve PSs, twenty PIs, seven PAs, and four PGs) and 49 SPs (eleven SMs, thirty-two Cers, and six Hexcers) (Figure 2C). Of these lipids, 127 species were detected in all samples, including thirty PEs, four LPCs, five LPEs, eleven PSs, twenty PIs, six PAs, four PGs, nine SMs, thirty-two Cers, and six Hexcers. In terms of numbers identified, PLs were easier to detect in NEG ion mode. Interestingly, PCs were only observed in POS ion mode because of the strong signal strength in this mode [40].

#### 3.4.2. Quantitative Analysis of Lipids

In POS ion mode, the total TG content of HM was 92.1% (Figure 3A). Interestingly, the value therein was slightly lower than that in earlier studies (98.68% and 95.13%) [14,15]. Freezing-thawing repeatedly as well as partial hydrolysis under the action of bile-salt-stimulated lipases in HM have been proven to decrease the content of TG [41,42]. Different analytical methods used in various studies can also be used to explain this discrepancy [14]. Meanwhile, the levels of DG decreased with increasing TG content. MG accounted for roughly 1% of all investigated samples. Except for GLs, the content of PLs was very low and did not differ among samples in POS ion mode. In this mode, PC (61.3%) was the most prevalent PL in HM, followed by PE (21.2%) (Figure 3A). However, Zhao et al. [14] demonstrated that SM (40.39%) was the major PLs type, followed by PE (27.39%) and PC (27.39%). Zou et al. [43] also observed SM as the most prevalent PL, followed by PC and PE. These discrepancies may be attributed to interindividual variation, detection technologies, and the use of standard and data processing methods [14]. It can be concluded that PC, PE, and SM are the main PLs in HM. As in HM, PC was the most prevalent PL type in formulas, accounting for over 65%. However, the level of PE was three times lower in IF1 (8.6%) and ten times lower in IF2 (2.4%) compared to that in HM (21.2%). In NEG ion mode, PE was the most abundant PL (39.4%) in HM, followed by Cer (19.4%) and SM (13.4%) (Figure 3A). On the contrary, the content of PE was two times lower in IF1 and a little higher in IF2 compared to that in HM. The levels of Cer and SM in IFs were lower than those in HM, probably resulting from the absence of MFGM [15]. These two types of lipids are involved in many physiological activities, for instance, cell growth, differentiation and withering [14]. Therefore, supplementation with MFGM to IFs may be an effective way to narrow the gap between human milk and formulas.

#### 3.4.3. Neutral Lipid Species Analysis

The content of a neutral lipid is shown in Appendix A. The carbon number (CN) and numbers of double bounds (DB) of the lipids in the samples ranged from 26 to 66 and from 0 to 13, respectively. Narrower ranges of lipids were reported by Tu et al. [9] in which CN and DB ranged from 30 to 60 and 0 to 10, respectively. Likely, Zhang et al. [10] found CN and DB in the ranges of 32 to 58 and 0 to 9, respectively. In HM, the CN 52 content of TGs was higher than the CN 54 content, while the opposite was observed in IFs [10]. This may be explained by the predominance of TG (18:1/16:0/18:1) (OPO) (5.46%) and TG (18:1/16:0/18:2) (OPL) (4.79%), which were the two major TG types in HM. Certainly, Zhao et al. [14] analyzed the lipid profiles of HM from four cities in China, and demonstrated the dominant place of OPO in milk lipid from Liuyang City and OPL from three other cities. OPO was also evidenced as the major TG type in HM from Spain, followed by OPL [44]. To the contrary, some researchers have declared that the predominant TG types in Chinese breast milk were OPL rather than OPO [7,15,45]. This may be explained by differences in diets or regions, as Chinese people have a higher intake of edible oils rich in LA, whereas olive oil containing more than 80% MUFAs is widely consumed in Western countries [14]. The other main TGs that accounted for over 3.5% were TG (10:0/12:0/18:1) (CaLaO) (3.73%), TG (12:0/12:0/18:2) (LaLaL) (3.60%), and TG (12:0/14:0/18:1) (LaMO) (3.56%), further verifying the predominance of OA and LA in HM. This went along with the above-mentioned results presented in Table 2. By contrast, the levels of OPO and OPL in formulas were far lower than those in HM. The content of OPO was roughly half of HM in IF1, and even lower in IF2. Despite the fact that the supplement of OPO was considered during the design of IF1, its content remained inadequate in comparison with HM. A similar pattern was also observed in OPL, which showed the much lower content in IF1 (2.58%) and IF2 (1.78%). Furthermore, the most abundant of TG types in IFs were TG (10:0/12:0/12:0) (CaLaLa) (8.01%, 10.94%), TG (10:0/10:0/12:0) (CaCaLa) (6.89%, 10.93%), and TG (12:0/12:0/14:0) (LaLaM) (6.12%, 8.44%), which presented at a lower level in HM. 

To sum up, as described in Figure 3B, TGs containing UFAs were the most abundant type in HM, while IFs were rich in TGs with three SFAs (SSS). Meanwhile, as shown in Figure 3C, TGs with two and three LCFAs (MLL, LLL) were the most plentiful TG type in HM, while IFs contained more medium-chain TGs (MCTs) (*p* < 0.05). Yuan et al. [46] showed that almost 30% of the total TGs were medium- and long-chain TGs (MLCTs), more than half were long-chain TGs (LCTs), and the levels of MCTs were less than 1% in HM. These characteristics of TGs will influence the digestion and absorption of lipids for infants. MCTs can be absorbed as TG and MCFAs and travel through the epithelium directly to the portal vein, providing energy rapidly [47]. In addition, in accordance with the levels of FAs, TGs with esterified OA were the major TG structures in HM (57.28% of total TGs), representing a significantly higher proportion than that in IFs (41.04% and 31.50%, respectively), and contained 72 TG types, followed by LA (49.10%, 24.67%, and 18.85%) with 55 TG types, and PA (37.32%, 25.77%, and 21.37%), with 38 types. Furthermore, TGs containing ARA and DHA in HM were at a higher prevalence than those in IFs, which was in accordance with the content of these two FAs discussed above. Intriguingly, we detected DG (16:0/18:1) and DG (18:1/18:2) (1.42% and 1.11%, respectively) as the most abundant DG species in HM, which were comparable with the milk fat reported by Zhao et al. [14]. However, some researchers identified DG (18:2/18:2) and DG (18:1/18:1) as the predominant DG types through analyzing lipid profiles of breast milk [7,15]. The major DGs in IF1 were same as those in HM, while DG (18:2/18:2) (0.84%) and DG (12:0/12:0) (0.62%) contributed the main DG types in IF2. MG (18:0) and MG (16:0) were the main two MG types observed in all these three investigated lipids. 

#### 3.4.4. Polar Lipid Species Analysis

The content of polar lipid is shown in Appendix A. It is generally acknowledged that PLs play irreplaceable roles in early brain development and cognitive ability for infants. 

In POS ion mode, PC (18:0/18:2) was the most abundant PC type identified in HM; a similar result was also reported previously [14]. However, lower levels of this type of PC were detected in formulas. Unlike what was found in HM, PC (16:0/18:1) and PC (18:2/18:2) were the predominant PC types in IFs. PCs are major sources of nerve choline, which is a precursor of the neurotransmitter acetylcholine, a regulator of signal transduction [14]. It has been reported that supplementation with PC is beneficial for guarding against gastrointestinal infections and diarrhea during early childhood [48]. In addition, PE (18:0/18:2) was the major PE type in all investigated milk samples in POS ion mode, and formulas had reduced levels of that compared to HM. Recently, Zhao et al. [14] also showed that the concentrations of PE (18:0/18:2) was the highest among all PEs in HM. In NEG ion mode, the most abundant of PE type was PE (18:0/18:1) (5.70%) in HM, and PE (18:0/18:2) in IF1 (3.69%) and PE (18:2/18:2) (9.94%) in IF2. Except as discussed above, PI, identified in NEG ion mode, was another major type of GPs accounted for 8.6% in HM, and 19.3% in IF1 and 31.7% in IF2. PI (18:1/18:1) (1.88%) was the most abundant PI type in HM, while PI (16:0/18:2) was the highest in formulas. 

In POS ion mode, we found SM (d16:0/22:0) to be the predominant SM type in both HM and formulas. Notably, all SMs identified here contained LCFAs (C14~C21) or very-long-chain FAs (≥C22) and were the main components of MFGM. This unique structure serves as a special point of adhesion for bacteria and viruses [7]. Meanwhile, in NEG ion mode, SM (d18:1/18:0) (5.70%) was the most abundant SM species in HM, although in IF1 and IF2, this place was occupied by SM (d18:1/16:0). Furthermore, the total level of SM in HM was higher than that in IFs. This may have been due to the absence of MFGM in IFs. SM is a basic component of myelin of the central nervous system and is involved in various physiological activities, functioning as the major source of nerve choline in the neonatal period [14]. Cer was another type of SPs detected in all milk samples. In POS ion mode, of the concentrations of Cer, (d18:0/14:0) was the highest Cer type in all samples. In NEG ion mode, the major type of Cer was Cer (d18:1/24:1) in HM. In comparison with HM, Cer (d18:1/22:0) and Cer (d18:1/23:0) occupied the first place of Cer type in IF1 and IF2, respectively.

### 3.5. Analysis of Differential Lipids 

First, the lipid species common to all milks (>1%) were visualized using heatmap analysis (Figure 4A,B). Obviously, HM showed plentiful amounts of TGs, with high CN (e.g., 52), whereas IFs were abundant in TGs with CN ≤ 38. Regarding PLs, the levels of PE 36:2 and PE 36:1 were higher in HM than in formulas, while the content of PE type with longer chain length showed the opposite. Lipids with significantly different abundances picked out by calculating P-Value among HM and formulas are shown in volcano plots (Figure 5A). In comparison with HM, the abundance of 63 differential lipids were downregulated and 80 species of lipids were upregulated in IF1. In IF2, 72 species of lipids were downregulated and 80 lipids were upregulated. Furthermore, there were some special lipid species detected in HM, which were seldom found in formulas (Appendix A). In detail, 25 species of lipids were detected in HM and absent for IF1, and the number was 23 in IF2. Different lipid profiles among HM and formulas contributed to distinct nutritive values, which will have profound and lasting impact on the health and development of infants [14].

To explore the relationships between differential lipids, SDL were screened using the screening criteria *p* < 0.001, VIP > 1, and fold change (FC, FC > 2 or <0.5) [22]. As shown in Figure 5B, a total of 67 SDLs were detected between HM and IF1, meanwhile, 73 SDLs were identified between HM and IF2. Heatmaps of SDLs were drawn and displayed in Figure 5C,D. These SDLs may act as biomarkers to identify HM and IFs. They also provide a direction for the design of IFs and related products. Further to study the quantitative interrelation of different lipid species visually, we performed the correlation network analysis of 67 and 73 SDLs on the basis of HM. As shown in Figure 5E,F, 134 correlations were identified for 67 SDLs and 138 correlations for 73SDLs (*p* < 0.05). Almost all types of lipids were correlated, especially within the same class. These strongly correlated lipids may serve similar biological function and nutritive value [22]. 

### 3.6. Pathway of Differential Lipids

In order to investigate the metabolic properties caused by structural differences of lipids, 12 subclasses, comprising 140 of the SDLs analyzed above between HM and formulas were mapped to the KEGG, HMDB, and PubChem databases. A total of 45 metabolic pathways were obtained when SDLs were searched against the KEGG pathway database (Appendix A). Furthermore, nine most relevant pathways were identified using MetaboAnalyst 5.0 and glycerophospholipid metabolism was the relevant metabolic pathway of the highest significance, followed by sphingolipid metabolism and glycerolipid metabolism (Figure 6). As discussed above, PCs and PEs were the most abundant GPs in HM. Given that PCs are essential for brain development and protect against inflammatory processes and PEs contribute to improving memory and enhancing brain function development [22,49], differences in the metabolic pathways in which SDLs participate may bring about the corresponding health effects to a certain extent.

## 4. Conclusions

We characterized and compared the lipids profiles from human milk and infant formulas using a quantitative lipidomics approach. HM was observed with much higher contents of total lipid and cholesterol, in comparison with IFs. Moreover, contrary to IFs, most SFAs were preferentially esterified at the *sn*-2 position in HM, while UFAs were preferentially esterified at the *sn*-1,3 position. In total, 425 lipid species assigned to 15 subclasses were characterized in HM and IFs. Among the 425 species, 222 and 127 common lipids were identified under the POS and NEG ion modes, respectively. HM contained more SM than formulas in both POS and NEG ion modes. Furthermore, 67 SDLs were identified between HM and IF1, and 73 SDLs were confirmed between HM and IF2. These SDLs might be used as indicators for distinguishing HM and IFs. In addition, we found the nine most relevant metabolic pathways, of which glycerophospholipid metabolism possessed the highest significance. However, this study was inadequate because of the low sample size, non-professional sample collection, and inconsistent maternal diets. More similar studies are warranted to deepen our understanding of the differences in lipid profiles among HM and IFs in the future. Overall, comprehensive analysis of the lipid profiles of HM and IFs will hopefully provide insights into the design of novel infant formulas that better mimic human milk.

## Figures and Tables

**Figure 1 foods-12-00600-f001:**
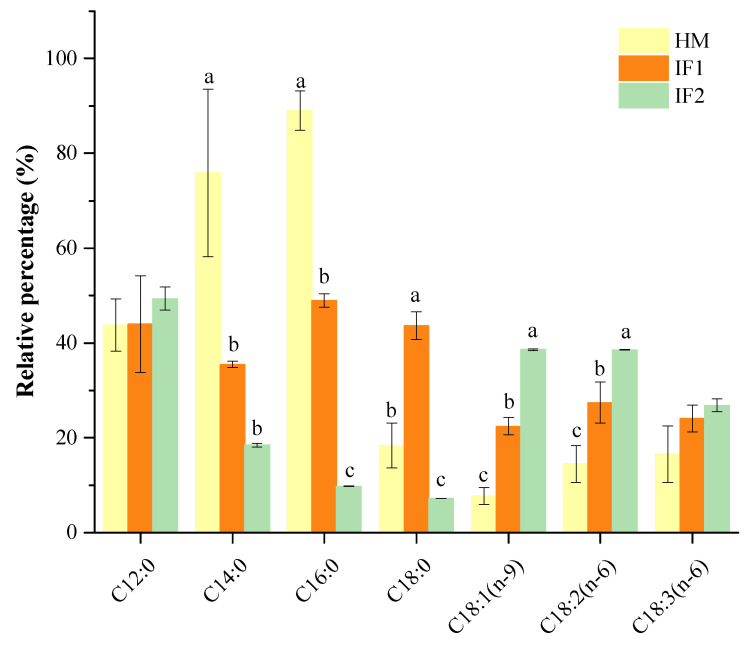
Percentage of relative *sn*-2 position fatty acids. Different lowercases indicate significant differences among samples. HM, human milk; IF, infant formula.

**Figure 2 foods-12-00600-f002:**
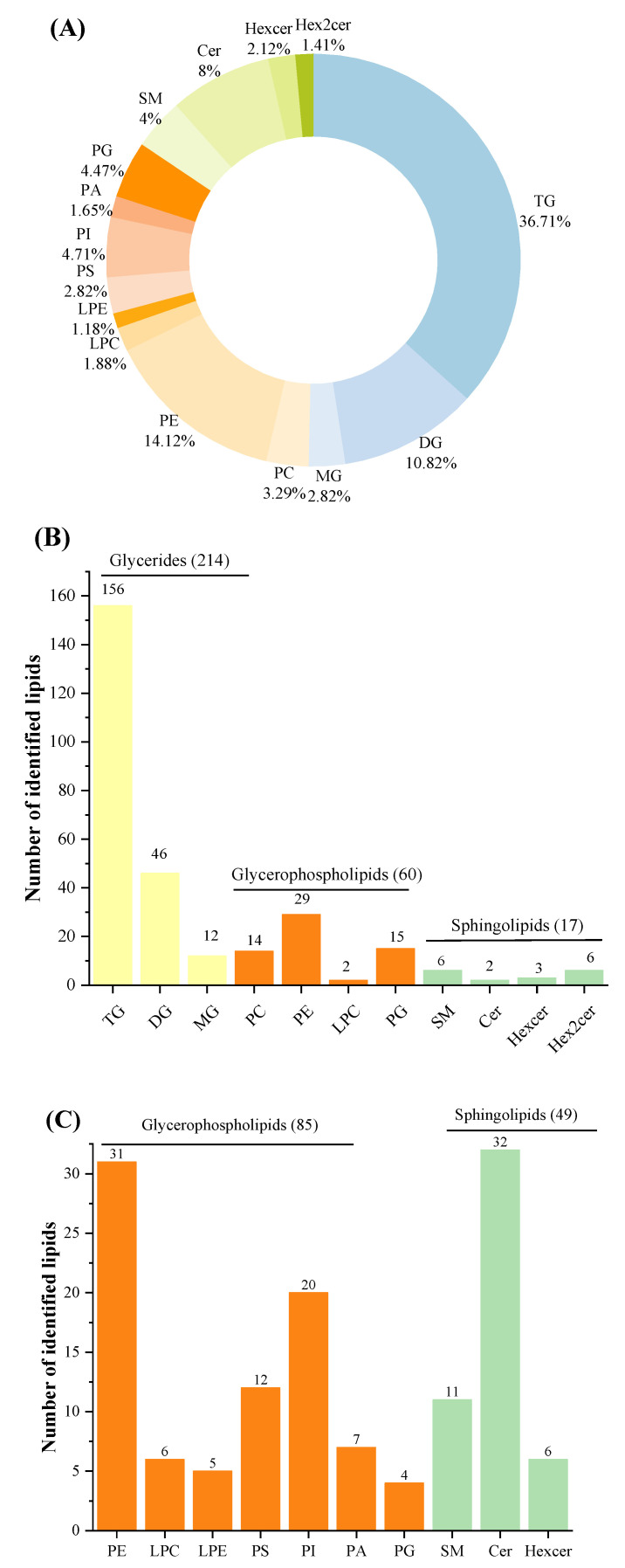
Total identified lipid subclasses in samples: percentages of numbers of lipid subclasses (**A**); numbers of each subclass identified in POS ion mode (**B**); and numbers of each subclass identified in Neg ion mode (**C**). POS, positive; NEG, negative.

**Figure 3 foods-12-00600-f003:**
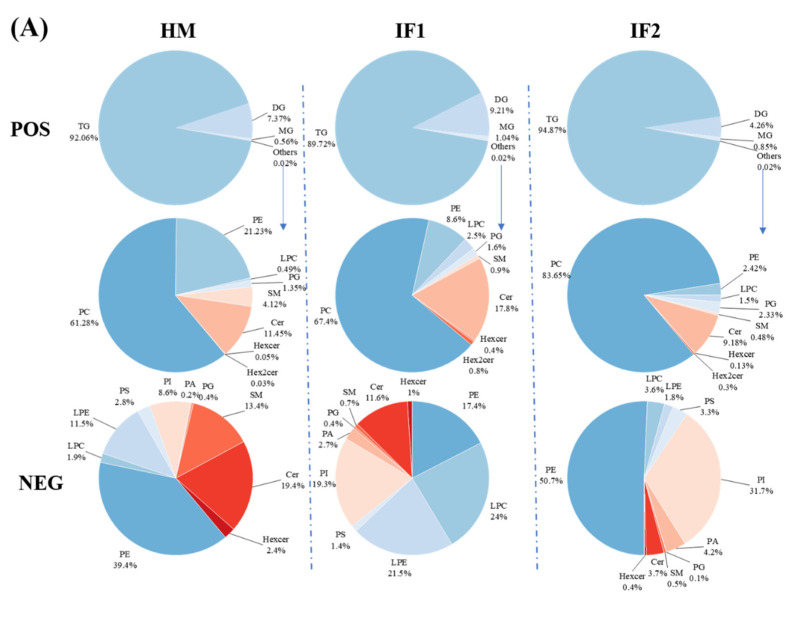
Comparison of percentage content of lipid subclasses among human milk and infant formulas in POS and NEG ion modes (**A**); the distribution of triglycerides in samples on the basis of saturation (**B**); and the distribution of triglycerides in samples on the basis of chain length (**C**). Different lowercase letters indicate significant differences among samples. MMM, three medium-chain fatty acids triglyceride; MML, two medium-chain and one long-chain fatty acids triglyceride; MLL, one medium-chain and two long-chain fatty acids triglyceride; LLL, three long-chain fatty acids triglyceride.

**Figure 4 foods-12-00600-f004:**
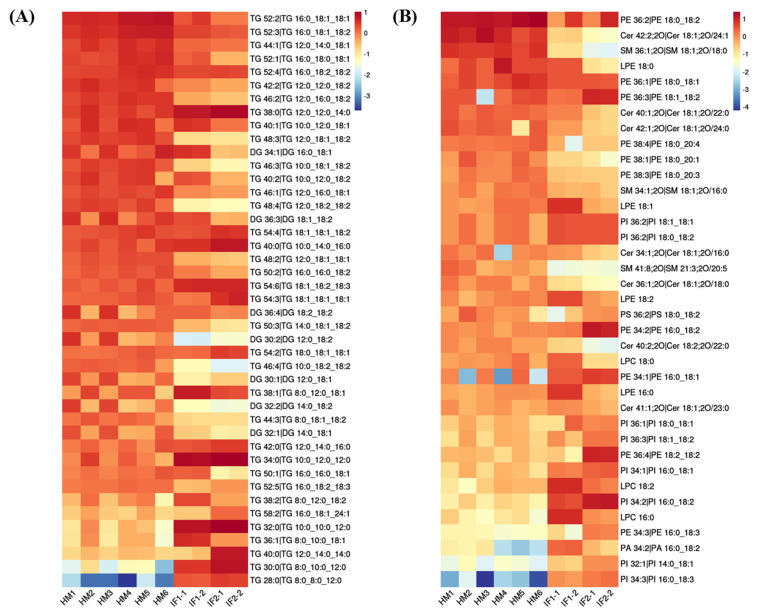
Heatmaps analysis of major common lipids in POS ion mode (**A**); and NEG ion mode (**B**).

**Figure 5 foods-12-00600-f005:**
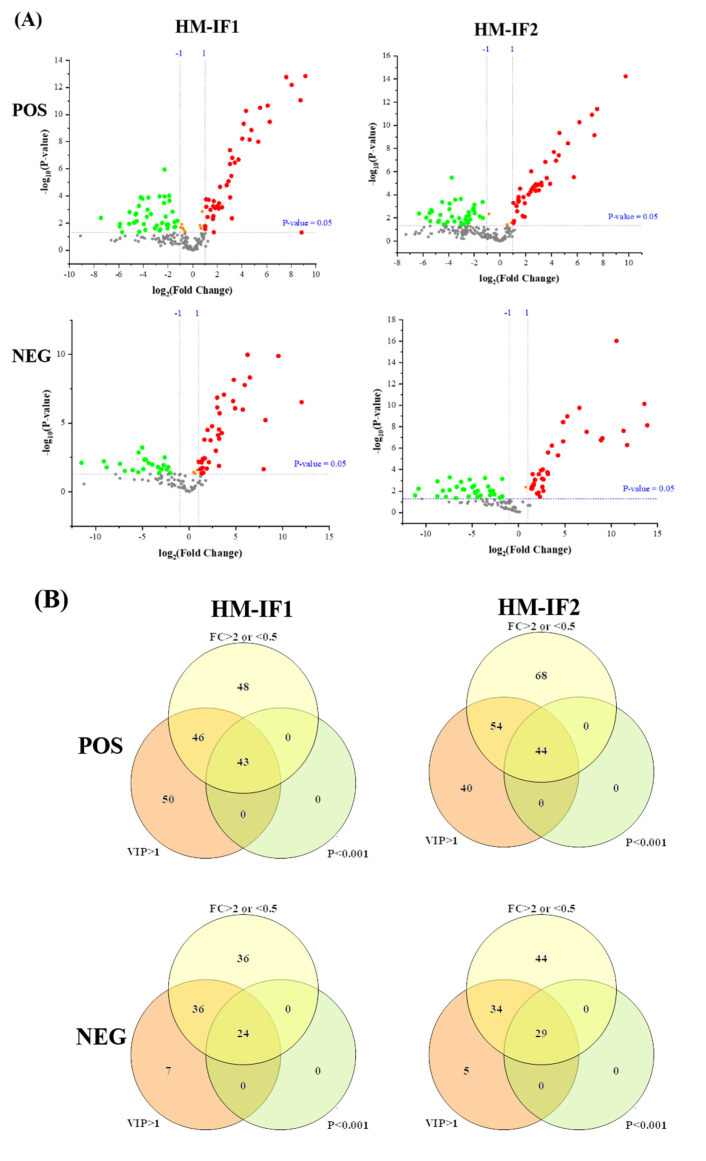
Volcano plot of differential lipids among HM and IFs (**A**). Red represents upregulated, and green represents downregulated. Venn diagrams of VIP, P-value, fold change results of HM and IFs (**B**); heatmap analysis of 67SDLs between HM and IF1 (**C**); 73 SDLs between HM and IF2 (**D**); correlation network (*p* < 0.05) of 67 SDLs in HM (**E**); and a correlation network of 73 SDLs in HM (**F**). The size of circle and width of line indicate the level of content and correlation. VIP, variable importance in projection; FC, fold change; P, P-value; SDL, significantly differential lipid; HM-IF1; comparison between HM and IF1; HM-IF2, comparison between HM and IF2.

**Figure 6 foods-12-00600-f006:**
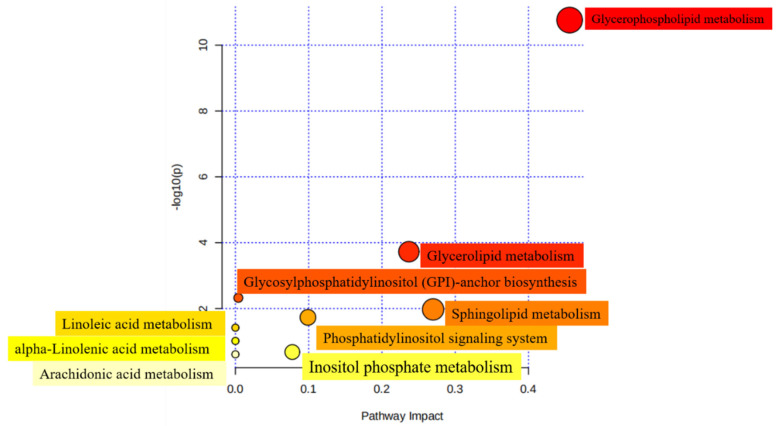
Metabolomic view of the significant lipid biosynthetic pathways in HM and IFs. Larger sizes and dark colors represent major pathway enrichment and high pathway impact values.

**Table 1 foods-12-00600-t001:** Content of lipid and cholesterol of human milk and infant formula.

Milk	Lipid Content (g/L)	Cholesterol Content (mg/100 g)	Percentage (%)
HM (*n* = 6)	43.90 ± 4.56 A	12.14 ± 3.09 a	0.277
IF1 (*n* = 2)	19.83 ± 0.13 B	2.87 ± 0.40 b	0.014
IF2 (*n* = 2)	21.79 ± 0.68 B	1.82 ± 0.17 b	0.008

The values are presented as means ± standard derivation. Different lowercase letters (a, b, and c) and uppercase letters (A, B and C) in the same column represent significant differences in content of cholesterol and lipid, respectively, among human milk and infant formulas (*p* < 0.05). Abbreviations are: HM, human milk; IF, infant formula; Percentage, (cholesterol content/lipid content).

**Table 2 foods-12-00600-t002:** Total fatty acids composition of human milk and infant formulas (%, *w*/*w*).

Fatty Acids	HM (*n* = 6)	IF1 (*n* = 2)	IF2 (*n* = 2)
C4:0	ND	0.02 ± 0.00	ND
C6:0	0.17 ± 0.07	0.07 ± 0.00	ND
C8:0	0.08 ± 0.03 c	0.24 ± 0.02 b	0.89 ± 0.08 a
C10:0	0.94 ± 0.04 b	0.51 ± 0.01 c	1.02 ± 0.02 a
C11:0	0.02 ± 0.01	ND	ND
C12:0	4.61 ± 0.78 b	1.79 ± 0.01 c	10.63 ± 0.40 a
C13:0	0.02 ± 0.02	ND	ND
C14:0	4.65 ± 0.99 ab	3.12 ± 0.04 b	5.13 ± 0.06 a
C14:1n-5	0.03 ± 0.02 b	0.16 ± 0.02 a	ND
C15:0	0.11 ± 0.05 b	0.28 ± 0.02 a	ND
C15:1n-5	0.02 ± 0.01	ND	ND
C16:0	20.92 ± 1.59 b	29.21 ± 0.16 a	10.08 ± 0.13 c
C16:1n-7	2.34 ± 0.63 a	0.41 ± 0.11 b	ND
C17:0	0.23 ± 0.09	0.16 ± 0.06	ND
C17:1n-7	0.08 ± 0.07	ND	ND
C18:0	6.05 ± 0.34 a	5.89 ± 0.08 a	4.31 ± 0.08 b
C18:1n-9 T	0.07 ± 0.07	ND	ND
C18:1n-9	35.56 ± 3.15 b	35.93 ± 0.10 b	45.64 ± 0.09 a
C18:1n-7	0.82 ± 0.15	ND	ND
C18:2n-6 T	0.01 ± 0.00	ND	ND
C18:2n-6	19.76 ± 3.65	19.03 ± 0.01	19.19 ± 0.03
C20:0	0.06 ± 0.05	ND	ND
C18:3n-6	1.39 ± 0.46	1.99 ± 0.01	1.67 ± 0.03
C20:1n-9	ND	0.21 ± 0.02	0.22 ± 0.06
C18:3n-3	0.51 ± 0.27	0.22 ± 0.08	0.18 ± 0.03
C20:2n-6	0.39 ± 0.14	ND	ND
C22:0	0.4 ± 0.11	ND	ND
C20:3n-6	0.4 ± 0.08 a	0.25 ± 0.02 b	0.37 ± 0.07 ab
C22:1n-9	0.02 ± 0.02 c	0.23 ± 0.01 b	0.48 ± 0.13 a
C20:4n-6	0.11 ± 0.11	ND	ND
C23:0	0.02 ± 0.01	ND	ND
C24:0	0.01 ± 0.00	ND	ND
C20:5n-3	ND	0.12 ± 0.06	ND
C24:1n-9	0.03 ± 0.02	ND	ND
C22:6n-3	0.27 ± 0.12	0.15 ± 0.06	0.17 ± 0.03
∑SFA	38.15 ± 0.82 b	41.28 ± 0.20 a	32.07 ± 0.35 c
∑UFA	61.7 ± 0.80 b	58.72 ± 0.20 c	67.93 ± 0.35 a
∑MUFA	38.97 ± 4.14 b	36.94 ± 0.26 b	46.34 ± 0.28 a
∑PUFA	22.84 ± 4.83 a	21.76 ± 0.24 a	21.58 ± 0.19 a
∑PUFA/MUFA	0.60 ± 0.16 a	0.59 ± 0.00 a	0.47 ± 0.00 a
∑n-6PUFA	22.05 ± 4.04 a	21.28 ± 0.01 a	21.23 ± 0.01 a
∑n-3PUFA	0.78 ± 0.28 a	0.49 ± 0.19 a	0.36 ± 0.06 a
SCFA	0.17 ± 0.07	0.08 ± 0.00	ND
MCFA (8~13)	5.66 ± 0.78 b	2.61 ± 0.00 c	12.55 ± 0.50 a
LCFA (14~21)	93.38 ± 0.71 b	96.98 ± 0.07 a	86.79 ± 0.34 c
Very long (≥22)	0.65 ± 0.28 a	0.39 ± 0.07 a	0.66 ± 0.16 a

The values are presented as means ± standard derivation. Different letters in the same row indicate significant differences among HM and IFs (*p* < 0.05). ND is not detected. Abbreviations are HM, human milk; IF, infant formula; T, trans; SFA, saturated fatty acids; UFA, unsaturated fatty acids; MUFA, monounsaturated fatty acids; PUFA, polyunsaturated fatty acids; SCFA, short-chain fatty acid; MCFA, medium-chain fatty acid; LCFA, long-chain fatty acid.

**Table 3 foods-12-00600-t003:** *sn*-2 fatty acids composition of human milk and infant formula (%, *w*/*w*).

Fatty Acids	HM (*n* = 6)	IF1 (*n* = 2)	IF2 (*n* = 2)
C4:0	ND	ND	0.04 ± 0.00
C6:0	0.04 ± 0.02 b	1.02 ± 0.21 a	0.11 ± 0.03 b
C8:0	0.02 ± 0.01 b	0.49 ± 0.31 a	0.05 ± 0.01 b
C10:0	0.43 ± 0.17	ND	0.41 ± 0.07
C12:0	6.06 ± 1.31 b	2.37 ± 0.56 c	15.85 ± 0.36 a
C14:0	10.34 ± 2.08 a	3.32 ± 0.1 b	2.85 ± 0.05 b
C15:0	0.31 ± 0.08	ND	ND
C16:0	55.77 ± 4.07 a	42.95 ± 1.01 b	2.98 ± 0.06 c
C16:1n-7	2.01 ± 1.00	ND	ND
C17:0	0.29 ± 0.08	ND	ND
C17:1n-7	0.14 ± 0.07	ND	ND
C18:0	3.3 ± 0.79 b	7.71 ± 0.40 a	0.95 ± 0.04 c
C18:1n-9	8.14 ± 1.94 c	24.22 ± 2.04 b	52.72 ± 0.38 a
C18:1n-7	0.24 ± 0.12	ND	ND
C18:2n-6	8.68 ± 3.38 b	15.67 ± 2.46 a	22.16 ± 0.15 a
C18:3n-6	0.7 ± 0.40 b	1.44 ± 0.17 a	1.37 ± 0.08 ab
C20:1n-9	0.16 ± 0.07	ND	ND
C18:3n-3	0.3 ± 0.12	ND	ND
C20:2n-6	0.15 ± 0.05	ND	ND
C22:0	0.11 ± 0.02	ND	ND
C20:3n-6	0.35 ± 0.18	0.82 ± 0.88	0.29 ± 0.09
C20:3n-3	0.58 ± 0.41	ND	ND
C20:4n-6	0.98 ± 0.65	ND	ND
C20:5n-3	0.16 ± 0.08	ND	ND
C22:5n-3	0.21 ± 0.06	ND	ND
C22:6n-3	0.27 ± 0.19	ND	0.23 ± 0.07
∑SFA	76.65 ± 5.79 a	57.85 ± 2.59 b	23.24 ± 0.62 c
∑MUFA	10.67 ± 2.97 c	24.22 ± 2.04 b	52.72 ± 0.38 a
∑PUFA	12.31 ± 3.40 b	17.93 ± 3.51 ab	24.05 ± 0.39 a

The values are presented as means ± standard derivation. Different letters in the same row indicate significant differences among HM and IFs (*p* < 0.05). ND is not detected. Abbreviations are HM, human milk; IF, infant formula; SFA, saturated fatty acids; UFA, unsaturated fatty acids; MUFA, monounsaturated fatty acids; PUFA, polyunsaturated fatty acids.

## Data Availability

Data is contained within the article.

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
