# Peer review of "Lipid Profiles of Human Milk and Infant Formulas: A Comparative Lipidomics Study"

_foods, 2023, doi:10.3390/foods12030600_

Round 1
Reviewer 1 Report
Journal: MDPI_foods
Title: Lipid profiles of human milk and infant formulas: A comparative lipidomics study
Manuscript ID: foods_2113076
Comments
This manuscript was studied to investigate the “Lipid profiles of human milk and infant formulas: A comparative lipidomics study”.
Authors adopted lipidomics to analyze and compare indepth lipid patterns of HM and IFs. Their results indicated that the distribution of fatty acids (FAs) and the structure of triacylglycerols were largely varied in samples analyzed. This study is reasonable and interesterified research field. Thus, this is a meaningful research in that it helps to undetstand the lipid profiles of human milk and infant formula.
However, some missing information must be worked out to make this article suitable for publication in MDPI_foods. Some of my suggestions are belows:
1. Line 87: Please explain the recommended intake stage of two infant formulas (IF1 and IF2). What is the definition of mature milk? Please compare the intake stage between infant formulas and human mature milks.
2. Line 111: Please explain the detailed infomation in Lipid extraction of Materials and Method. For example, The concentration of IS mixture, the dissolved solvent, and etc.
3. Line 119: Please suggest the reference in gravimetric method for the determination of lipid content.
4. Line 120-126: In HPLC analysis of cholesterol, cutoff value of Methanol is 205 and is it possible to analyze in 205 nm of detector? Please check it or present the chromatogram.
- Line 124: Please edit it with “SPD-20A UV/VIS detector” instead of “SPD-20A detector”
5. The lipid composition of human milk is closely related to maternal dietary food intake. Can the samples colleced in this study be representative of breast human milk?
6. Line 144: Please suggest the target compoents of lipid profiles such as PC, PE, LPC, LPE, Cer, Hexcer, and etc. In this lipidomics study, the lipid profiles are important values. Therefore, the extact detailed analysis methods should be suggested in Materials and Methods. Compared to the analysis data, the description of the analysis method is insufficient.
Reviewer 2 Report
I have indicated all my comments in the pdf version of the manuscript and which is attached in this field

Author Response
Please see the attchment

Round 2
Reviewer 2 Report
Reviewer 2 Comments
Point 24: How do you establish a statistical difference with a fatty acid that youd could not detect? Response 24: We used “0” for the ones that are not detected. Then a statistical analysis was performed.
New comment:
I suggest deleting all the statistical analysis involving non-detected (ND) samples. The authors should keep in mind that ND is not equivalent to zero value. Consequently, a statistical analysis with a ND value does not make much sense.
Point 25: “……In detail, the former presented SFA content as much as three times higher than that in IF2 (23.24%).……” The sentence is grammatically incorrect. It is hard to digest
Response 25: Thanks for your comment. The sentence has been changed to “In detail, the concentration of SFAs in HM (76.65%) was even three times higher than that in IF2 (23.24%)” at line 294 (page 8).
New comment:
I suggest rephrasing as “More specifically, the SFAs concentration in HM (76.65%) was 3.3 times higher than in IF2 (23.24%)”
Point 38: “To explore the relationships between differential lipids, SDL were screened using the screening criteria P < 0.001, VIP >1, and fold change (FC, FC >2 or < 0.5) [22]. As shown in Figure 5B, a total of 67 SDLs was detected between HM and IF1, meanwhile, 73 SDLs were identified between HM and IF2. Heatmaps of SDLs were drawn and displayed in Figure 5C-D. These SDLs may act as biomarkers to identify HM and IFs. They also provide a direction for the design of IFs and related products. Further to study the quantitative interrelation of different lipid species visually, we performed the correlation network analysis of 67 and 73 SDLs on the basis of HM……” How reliable is this VIP study. Unfortunately, the number of declared samples is too low. Maybe you should explain the grounds that confer to your study the require statistical power to validate your conclusions
Response 38: Thanks for your comments. We fully understand your question. The samples size in this study is insufficient. The difficulty of collecting samples is one of the important problems because of the rarity and variability of breast milk. To solve this problem, we have done our best to collect a representative sample of human mature milk (30~32 days). We try to ensure the uniformity of the donor's dietary pattern, age, gestational age, physical condition, etc. Also, the collection time was paid attention to. In addition, more stringent criterion (P<0.001, VIP>1, and FC>2 or <0.5) were used when differential lipid analysis is performed on breast milk and infant formula. Of course, the larger the sample size, the more reliable the results. We need to improve the reliability of the conclusion in the future research.
New comment:
Maybe the authors should mention in the conclusion or in another section some of the weaknesses of the present study (e.g. lack of statistical power due to the low number of analyzed samples) in that way it could be an aspect that potential readers interested in performing a similar research could take into account.
Author Response
Point 24: How do you establish a statistical difference with a fatty acid that youd could not detect?
Response 24: We used “0” for the ones that are not detected. Then a statistical analysis was performed.
Comment 1: I suggest deleting all the statistical analysis involving non-detected (ND) samples. The authors should keep in mind that ND is not equivalent to zero value. Consequently, a statistical analysis with a ND value does not make much sense.
Response 1: Thanks for your comment. As you suggested, we have deleted all the statistical analysis involving non-detected (ND) samples in the revised version (line 237, page 5; line 287, page7).
Point 25: “……In detail, the former presented SFA content as much as three times higher than that in IF2 (23.24%).……” The sentence is grammatically incorrect. It is hard to digest
Response 25: Thanks for your comment. The sentence has been changed to “In detail, the concentration of SFAs in HM (76.65%) was even three times higher than that in IF2 (23.24%)” at line 294 (page 8).
Comment 2: I suggest rephrasing as “More specifically, the SFAs concentration in HM (76.65%) was 3.3 times higher than in IF2 (23.24%)”
Response 2: Thanks for your suggestion. The sentence has been changed to “More specifically, the SFAs concentration in HM (76.65%) was 3.3 times higher than in IF2 (23.24%)” in the revised version at line 293 (page8).
Point 38: “To explore the relationships between differential lipids, SDL were screened using the screening criteria P < 0.001, VIP >1, and fold change (FC, FC >2 or < 0.5) [22]. As shown in Figure 5B, a total of 67 SDLs was detected between HM and IF1, meanwhile, 73 SDLs were identified between HM and IF2. Heatmaps of SDLs were drawn and displayed in Figure 5C-D. These SDLs may act as biomarkers to identify HM and IFs. They also provide a direction for the design of IFs and related products. Further to study the quantitative interrelation of different lipid species visually, we performed the correlation network analysis of 67 and 73 SDLs on the basis of HM……” How reliable is this VIP study. Unfortunately, the number of declared samples is too low. Maybe you should explain the grounds that confer to your study the require statistical power to validate your conclusions
Response 38: Thanks for your comments. We fully understand your question. The samples size in this study is insufficient. The difficulty of collecting samples is one of the important problems because of the rarity and variability of breast milk. To solve this problem, we have done our best to collect a representative sample of human mature milk (30~32 days). We try to ensure the uniformity of the donor's dietary pattern, age, gestational age, physical condition, etc. Also, the collection time was paid attention to. In addition, more stringent criterion (P<0.001, VIP>1, and FC>2 or <0.5) were used when differential lipid analysis is performed on breast milk and infant formula. Of course, the larger the sample size, the more reliable the results. We need to improve the reliability of the conclusion in the future research.
Comment 3: Maybe the authors should mention in the conclusion or in another section some of the weaknesses of the present study (e.g. lack of statistical power due to the low number of analyzed samples) in that way it could be an aspect that potential readers interested in performing a similar research could take into account.
Response 3: Thanks for your comment. We have supplemented the weakness of the present study in the conclusion at line 542 (page 19).